# Quantitative in vivo whole genome motility screen reveals novel therapeutic targets to block cancer metastasis

Konstantin Stoletov[1], Lian Willetts[1], Robert J. Paproski[1], David J. Bond[1], Srijan Raha[1], Juan Jovel[2,3], Benjamin Adam[4], Amy E. Robertson[1], Francis Wong[1], Emma Woolner[1], Deborah L. Sosnowski[1], Tarek A. Bismar[5], Gane Ka-Shu Wong[2,3,6], Andries Zijlstra [7] & John D. Lewis [1]

Metastasis is the most lethal aspect of cancer, yet current therapeutic strategies do not target its key rate-limiting steps. We have previously shown that the entry of cancer cells into the blood stream, or intravasation, is highly dependent upon in vivo cancer cell motility, making it an attractive therapeutic target. To systemically identify genes required for tumor cell motility in an in vivo tumor microenvironment, we established a novel quantitative in vivo screening platform based on intravital imaging of human cancer metastasis in ex ovo avian embryos. Utilizing this platform to screen a genome-wide shRNA library, we identified a panel of novel genes whose function is required for productive cancer cell motility in vivo, and whose expression is closely associated with metastatic risk in human cancers. The RNAi-mediated inhibition of these gene targets resulted in a nearly total (>99.5%) block of spontaneous cancer metastasis in vivo.

[1] Department of Oncology, University of Alberta, Edmonton, AB T6G 2E1, Canada. [2] Department of Biological Sciences, University of Alberta, Edmonton, AB T6G 2E9, Canada. [3] Department of Medicine, University of Alberta, Edmonton, AB T6G 2E1, Canada. [4] Department of Laboratory Medicine and Pathology, University of Alberta, Edmonton, AB T6G 2E1, Canada. [5] Departments of Pathology and Laboratory Medicine, Oncology, Biochemistry and Molecular Biology, University of Calgary Cumming School of Medicine and Calgary Laboratory Services, Calgary, AB T2V 1P9, Canada. [6] BGI-Shenzhen, Beishan Industrial Zone, Yantian District, 518083 Shenzhen, China. [7] Department of Pathology, Microbiology and Immunology, Vanderbilt University, 1161 21st Ave. S., C-2104A MCN, Nashville, TN 37232-2561, USA. These authors contributed equally: Konstantin Stoletov, Lian Willetts. Correspondence and requests for materials should be addressed to J.D.L. (email: jdlewis@ualberta.ca)

Metastatic dissemination is the primary cause of cancer-related deaths[1–4]. While surgical resection of primary tumors in concert with systemic chemotherapy has provided success in the treatment of localized cancers, metastatic disease has proven remarkably resistant to modern targeted therapies, rendering these cancers incurable. Indeed, to mitigate the risk of future metastasis, many patients are subjected to highly morbid treatment regimens that negatively impact quality of life[5]. Therapies that specifically target the rate-limiting steps of metastatic dissemination of tumor cells could significantly improve cancer treatment by removing the threat of systemic disease and decreasing our dependency on systemic therapies with detrimental side-effects[1–4].

The process of metastasis is dependent on a tumor cell's ability to intravasate into the blood stream, disseminate to a distant site, evade immune detection, survive, proliferate and subsequently colonize a new microenvironment[6]. Previously, we have shown that intravasation rates are highly dependent on in vivo tumor cell motility. Furthermore, when motility is inhibited using a migration-blocking antibody that targets tetraspanin CD151, both cancer cell intravasation and distant metastasis are blocked[3,7]. Given that the genes and signaling networks that drive in vivo motility and intravasation are different from those required for efficient primary tumor formation, we sought to develop an in vivo approach to feasibly screen for genes required for motility, and thus intravasation and metastasis[8].

Previously, the identification of genes required for in vivo cell motility has been impeded by the inherent difficulty in visualizing the formation of metastatic lesions in vivo[9,10]. To address this, we utilized a novel intravital imaging approach in shell-less, ex ovo avian embryos to perform an shRNA screen for gene products that regulate tumor cell motility in vivo[11,12]. Here, we describe the discovery of novel genes that drive cancer cell motility and metastasis in vivo. We show that targeting of these genes blocks productive cancer cell invasion and inhibits spontaneous metastasis in a mouse model of human cancer progression. The expression of these genes positively correlates with progression of several human cancers, highlighting their promise as therapeutic targets.

## Results

**Visualizing cancer cell motility phenotypes in the avian embryo.** Upon intravenous injection into the avian embryo, cancer cells disseminate throughout the vasculature. A substantial fraction of these cancer cells arrest as single cells in the chorioallantoic membrane (CAM), where they undergo extravasation into the extravascular stroma and proliferate into invasive metastatic colonies[13]. These colonies, each derived from a single cancer cell, reach the size of ~1 mm$^2$ (50−100 cells per colony) over 4 days and can be easily visualized using intravital microscopy (Fig. 1a and Supplementary Fig. 1a, b). Because thousands of individual metastatic colonies can be simultaneously visualized in the CAM of a single embryo, it is feasible to screen large libraries of genes using this approach. When highly motile cancer cells such as the human head and neck HEp3 cell line are injected, the resulting colonies adopt a diffuse "spread out" morphology where the proliferating cells have migrated a significant distance from the point of extravasation (Supplementary Fig. 1b). When the in vivo motility of tumor cells is reduced, such as that observed when using a CD151-specific migration-blocking antibody, metastatic colonies exhibit a highly compact morphology that is easily distinguished from the highly motile phenotype[3]. These compact metastatic lesions, comprised of tightly packed cancer cells, can be readily excised from the surrounding tissue and subjected to further analysis. We hypothesized that, as we

had seen with the targeting of CD151, the inhibition of genes required for in vivo cell motility would lead to compact colony phenotypes, thereby allowing us to utilize this approach to screen for therapeutic targets of cell motility that would in turn impact intravasation and metastasis.

**Intravital imaging screen for genes required for productive motility.** To perform the screen, we transduced HEp3 cells with a human shRNAGIPZ microRNA-adapted shRNA lentiviral library (Open Biosystems) built using a native miR-30 primary transcript to enable processing by the endogenous RNAi pathway. This library contains 79,805 sequence-verified shRNAs targeting 30,728 human genes contained in 7 pools, along with TurboGFP to monitor successful transduction. Each pool was used to transduce HEp3 cells in culture at an MOI (0.2), favoring a single shRNA integration per cancer cell according to Poisson distribution. When 25,000 tumor cells are injected intravenously into the avian embryo, roughly 10% of the cells arrest and extravasate in the visible and accessible CAM to form isolated metastatic colonies (Fig. 1a)[3]. To ensure 3× coverage of the 79805 shRNA clones with 99% confidence, the screen was performed in 100 embryos. Transduced GFP-expressing cells were injected intravenously into embryos in ex ovo culture at developmental day 10. On developmental day 15, more than 200,000 colonies in the CAMs of the 100 embryos were surveyed using intravital microscopy. Of these, 67 morphologically compact metastatic lesions were identified and excised. These colonies were dissociated and cultured under selection, and 50 clones were successfully expanded in culture.

To identify the integrated shRNA, inserts from each clone were amplified by PCR using common flanking primers and the resulting cDNA sequences were determined by deep sequencing on an Illumina platform. Raw sequence reads were subjected to a stringent filtering algorithm to identify the flanking miRNA sequences and exclude reads with inconsistent loop sequences and stem base-pair mismatches. Filtered sequences were then subjected to BLAST analysis against both the library and the human nucleotide (nt) database and ranked according to abundance (Fig. 1a and Supplementary Data 1). We found that 17 of the 50 isolated clones contained a single shRNA, while the remaining 33 clones each contained more than one shRNA (Supplementary Data 1).

**Identified genes are required for productive cancer cell invasion in vivo.** The gene targets were then prioritized based on their impact on productive cell migration in vivo according to the degree of their compact colony phenotype. The phenotype of each clone was validated using an experimental metastasis approach. Clones were injected intravenously into ex ovo chicken embryos and images of the resulting metastatic colonies were captured using intravital imaging. We developed a custom MATLAB-based program to analyze the images of each metastatic colony using three complementary algorithms (Fig. 1 and Supplementary Fig. 1). While we did not detect significant differences in the rate of proliferation of the clones in vitro, we observed that several clones grew at different rates in vivo (Supplementary Fig 2a). Therefore, to mitigate the effect of differences in proliferation between individual colonies and to get an accurate assessment of in vivo cancer cell motility, we designed algorithms to analyze three distinct parameters: (A) cancer cell remoteness from the colony centroid (Linear index); (B) the density of cancer cells within the metastatic colony area (Density index) and; (C) the total area occupied by each metastatic colony (Area index, Supplementary Figs. 1, 2). Briefly, the first algorithm creates a mask using fluorescence to delineate the cancer cells and

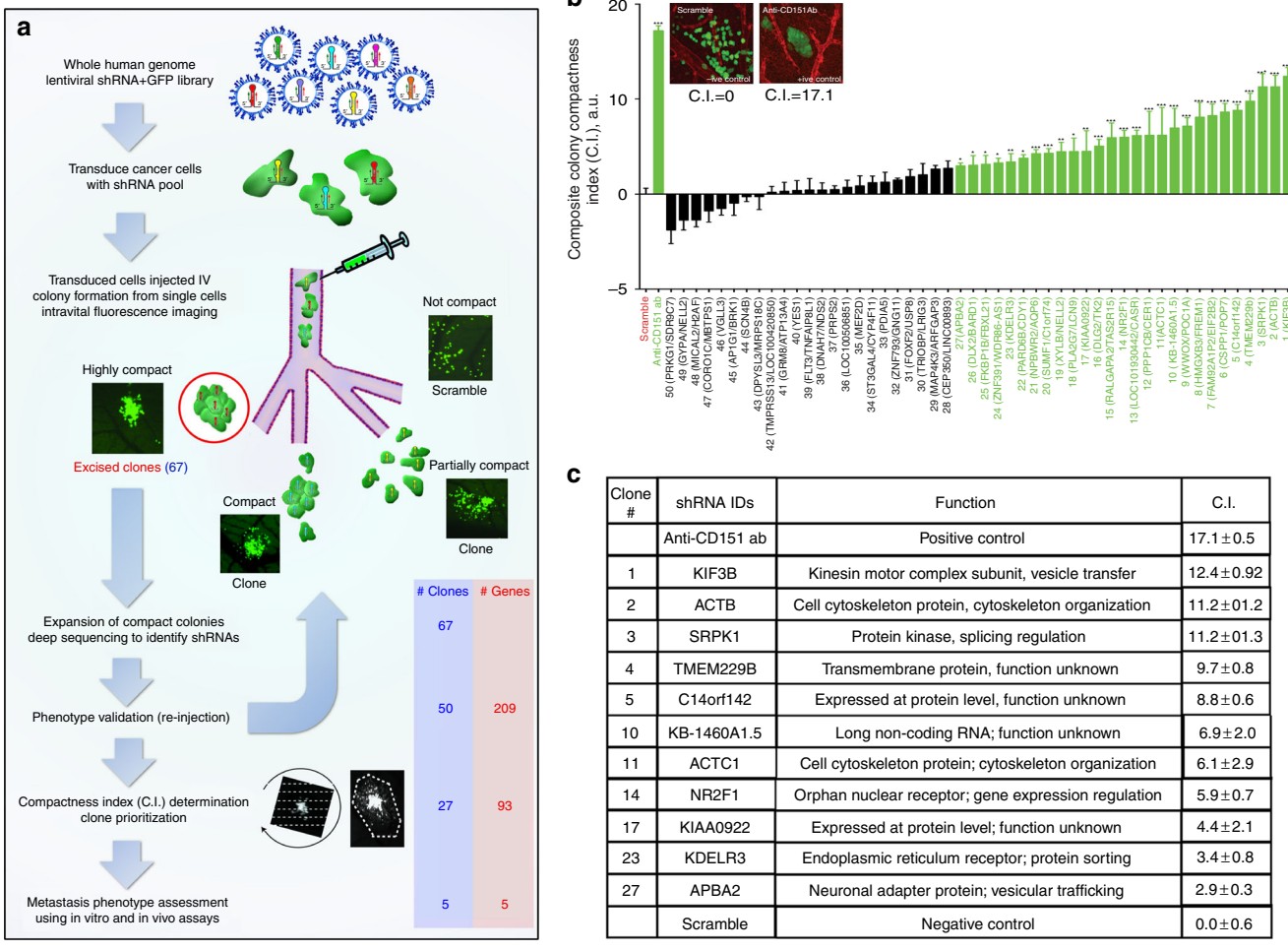

**Fig. 1** Overview of the in vivo screen for genes required for productive motility. **a** HEp3 cells were transduced with a pooled whole human genome lentiviral shRNA library and injected intravenously into 100 ex ovo avian embryos. Compact metastatic colonies derived from single cancer cells were excised 6 days post injection, expanded and analyzed by Illumina deep sequencing. Colonies were re-injected individually to validate their phenotype and prioritized based on a composite compactness (C.I.) index. Selected screen hits that produced metastatic colonies that were significantly more compact than those produced by cancer cells transduced with scramble shRNA transduced cells were selected for further analysis. **b** Composite compactness index (C.I.) distribution of screen hits relative to positive (anti-CD151) and negative (scramble shRNA) controls. Screen hits that are significantly more compact than negative control are indicated in green. Clones containing a single shRNA species are in bold. For clones containing multiple shRNAs, the two most predominant shRNAs are shown. Statistical significance was determined using one-way ANOVA with Fisher's LSD test (*$p < 0.05$, **$p < 0.01$, ***$p < 0.001$). **c** Table summarizing shRNA gene IDs from significantly compact clones containing single shRNA. Gene cards (http://www.genecards.org) and KEGG pathway (http://www.genome.jp) databases were used for gene function annotations

uses a 360° line-scan through the centroid to build an average line plot fitted to a Gaussian distribution (Supplementary Fig. 1c). The deviation in Gaussian radial line-scan intensity distribution between colonies formed by individual clones relative to control shRNA colonies is used to generate the Linear index. The second and third algorithms use the fluorescence mask to measure individual metastatic colony areas (Area Index) and calculate the fluorescence density within each area (Density index) (Supplementary Fig. 1d). A minimum of ten individual colonies for each clone were analyzed. While each index produced a similar ranking of the colonies identified in the screen, each method poorly identified a number of visually compact clones when used alone (Supplementary Fig. 2b−d). For this reason, the three algorithms were combined to create a composite colony Compactness Index (C.I.) that was used to stratify the phenotypes of the hit clones compared to the anti-CD151 antibody-treated positive control and the scrambled shRNA negative control

(Supplementary Data 1, Fig. 1b and Supplementary Fig. 2). The C.I. was calculated from the Z-scores (experimental − control / SD control) for each Index where C.I. = Z(Density Index) − Z(Linear Index) − Z(Area Index).

The morphology of positive control colonies, generated after treatment with the CD151-targeted migration-blocking antibody (positive control), exhibited the most dramatic increase in C.I. (17.1 ± 1.68) compared to highly invasive metastatic colonies generated by negative control cells expressing scramble shRNA (negative control, 0 ± 0.6) (Fig. 1b, Supplementary Fig. 2a). Statistical analysis of the C.I. index revealed 27 clones with metastatic colony phenotypes whose C.I. differed significantly ($p \leq 0.05$) from those of the negative control (Fig. 1b). Eleven (11) out of these 27 clones contained single shRNAs (KIF3B, ACTB, SRPK1, TMEM229B, C14orf142, KB-1460A1.5, ACTC1, NR2F1, KIAA0922, KDELR3, and APBA2). Clones containing a single shRNA and C.I. ≥ 5.0 were selected for downstream analysis (Fig. 1c).

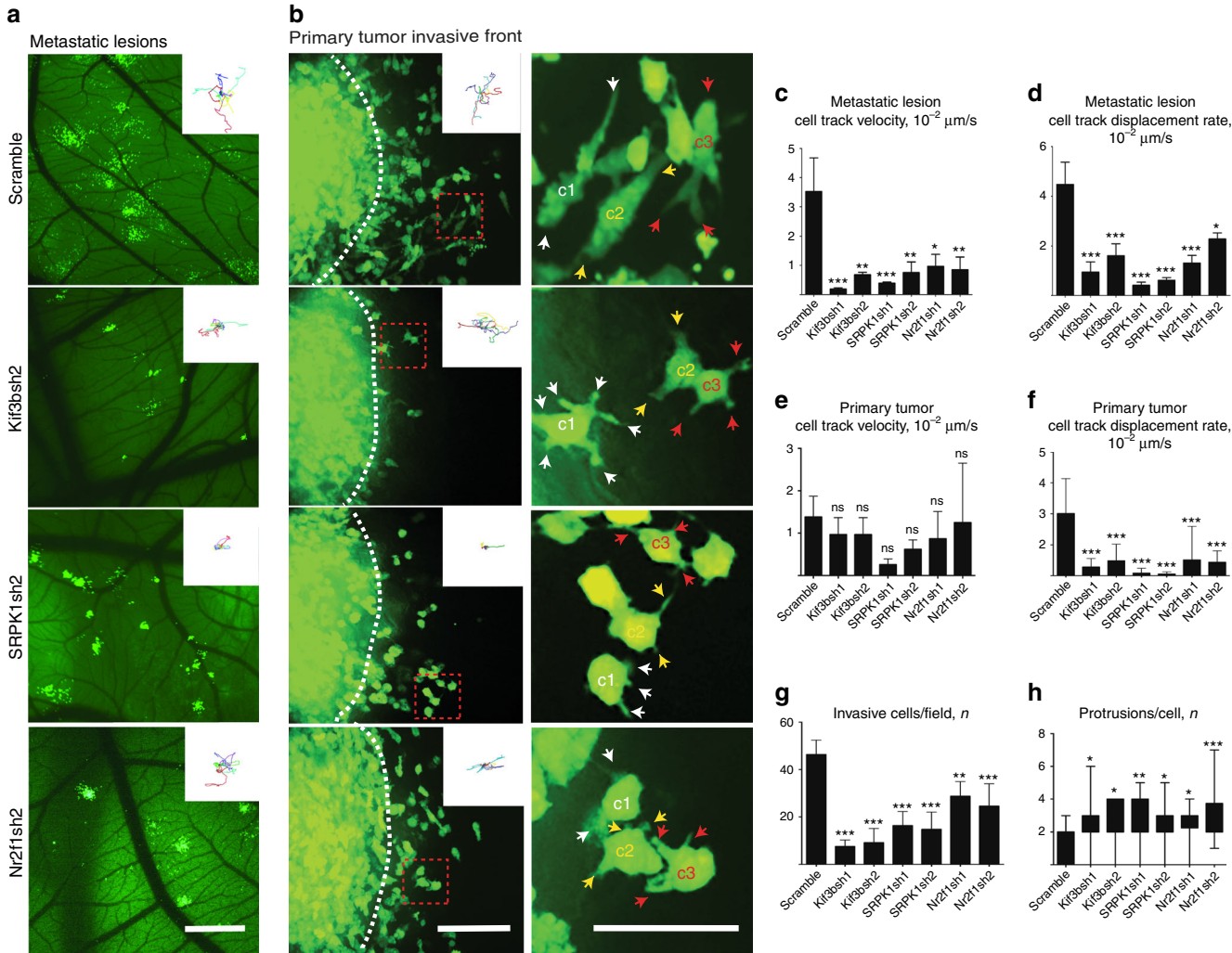

**Fig. 2** Screen-identified genes are required for productive cancer cell invasion in vivo. An intravital imaging approach was utilized to visualize and quantify the behavior of cancer cells in metastatic colonies and primary tumors over 7 or more hours. **a** Metastatic colonies arising from single HEp3 cells transduced by scramble shRNA or shRNAs targeting KIF3B, SRPK1, or NR2F1. Insets show representative cell tracks within the metastatic colonies. **b** Visualizing the invasive front of primary tumors (left panel) produced by HEp3 cells transduced by scramble shRNA or shRNAs targeting KIF3B, SRPK1, or NR2F1. Insets show representative cell tracks at the invasive fronts. Right panel shows a close-up of the cells from red dashed squares in the left panel. Color-coded arrows point to cell protrusions formed by the individual, correspondingly color-coded labeled cells (c1−c3). **c** Average in vivo cancer cell migration velocity for control and knockdown clones in metastatic colonies. **d** Average in vivo cell displacement rate (productive migration) for control and knockdown clones in metastatic colonies. **e** Average in vivo cancer cell migration velocity for control and knockdown clones in the invasive front of primary tumors. **f** Average in vivo cell displacement rate (productive migration) for control and knockdown clones in the invasive front of primary tumors. **g** Average number of invasive cells per field that migrated beyond the primary tumor periphery. **h** Average number of protrusions per cell for control and knockdown clones. Scale bars = 500 µm (**a**); 200 µm (**b**, left panel) or 20 µm (**b**, right panel)

To confirm that the observed inhibition of in vivo motility was due to the shRNA-mediated depletion of the target gene(s) and not an off-target effect, we utilized independent shRNA constructs to create new HEp3 clones for KIF3B (C.I. = 12.4), SRPK1 (C.I. = 11.2), TMEM229B (C.I. = 9.7), C14orf142 (C.I. = 8.8), and NR2F1 (C.I. = 5.9). Comparison of the target gene and protein expression in the original and newly derived clones confirmed the efficient knockdown of the targets in each cell line (Supplementary Fig. 3a−e). The clones bearing independent shRNAs (designated sh2) were then validated using the in vivo metastatic colony formation assay. All clones reproduced the compact colony phenotype with C.I. values similar to those of their primary screen hit clone (Supplementary Fig. 3f).

To gain additional insight into the migratory phenotypes induced by knockdown of these genes, we performed high-resolution in vivo time-lapse imaging of individual metastatic colonies and the invasive front of primary tumors derived from each clone and the control (scramble) shRNA transduced HEp3 cancer cells. For these studies, we concentrated our efforts on two clones with high C.I. (KIF3B and SRPK1) and one with lower C.I. (NR2F1). We observed that shRNA-mediated inhibition of each of these targets significantly reduced the velocity and productivity (the net straight-line displacement of a cell from its original position per unit time) of cancer cell migration in vivo (Fig. 2a−f, and Supplementary Movies 1, 2). Cancer cells from each of the KIF3Bsh1/sh2, SRPK1sh1/sh2, and NR2F1sh1/sh2 clones displayed significantly reduced motility and productive migration in

metastatic colonies (Supplementary Movie 1 and Fig. 2a, c, d). While there was no significant reduction in the average velocity of cancer cells at the invasive front of the primary tumor, productive motility at the invasive front was reduced by more than 75% in all clones (Supplementary Movie 2 and Fig. 2b, e, f). This corresponded well with the significant reduction in the number of invasive cancer cells observed in the invasive zone around primary tumors derived from KIF3Bsh1/sh2, SRPK1sh1/sh2, and NR2F1sh1/sh2 clones compared to the control (Fig. 2g). Phenotypically, scramble shRNA control HEp3 cells at the invasive front tended to form single dominant protrusions in the direction of motility while KIF3Bsh1/sh2, SRPK1sh1/sh2, and NR2F1sh1/sh2 clones formed multiple protrusions extending in all directions in an uncoordinated fashion (Fig. 2b, h and Supplementary Movies 1, 2).

**Kif3b regulates the interaction of cancer cells with the extracellular matrix.** Cancer cells invade tissues using the guidance of tissue structural elements such as vasculature and the collagen-rich extracellular matrix[14–17]. Highly metastatic tumor cells display increased affinity to the vasculature (vasculotropism) and ability to invade into and rearrange collagen tissue matrix[14–17]. Structurally, the chicken embryo CAM is a transparent organ roughly 200 μm thick consisting of a vascular network surrounded by a dense collagen-rich matrix that can be visualized in total using in vivo multiphoton imaging (Fig. 3a). Structurally, it bears a high degree of similarity with mouse lung (Fig. 3b), and human cancer cells appear to robustly interact with the collagen fiber network in both tissues (Fig. 3b). We investigated the role of top target Kif3b in this microenvironment using multiphoton imaging with second harmonic generation (SHG) to visualize collagen. We found that control HEp3 cells robustly spread within the CAM vasculature/collagen fiber network with many cells directly attaching to the vascular walls and forming long-lived dominant protrusions that frequently extend along the individual collagen fibers (Fig. 3c, e–g and Supplementary Movies 3, 4). In contrast, we found that mutant cells that were engineered using Kif3B targeting shRNAs showed decreased vasculotropism and were often surrounded by areas of low collagen density (Fig. 3 and Supplementary Movies 3, 4). Cell protrusions formed by KIF3B mutant cells rarely engaged collagen fibers and displayed significantly shorter lifetime (Fig. 3c–g, and Supplementary Movies 3, 4). Both shRNAs targeting Kif3b protein expression showed very similar effect on HEp3 cancer cell interaction with the vasculature and collagen fiber network (Figs. 3 and 4 and Supplementary Figure 4). This phenotype was also observed in KIF3B mutant human HT1080 fibrosarcoma and MDA231 breast cancer cells (Supplementary Fig. 3a and Supplementary Fig. 5a–g).

We then examined the cancer cell−collagen matrix interaction at the invasive primary tumor front. Invading tumor cells actively reorganize the collagen-rich matrix at primary tumor front creating areas of densely bundled, aligned collagen fibers that are used as pathways for invasion out of the primary tumor[16,17]. Indeed, HEp3 control tumor fronts had numerous areas of collagen fiber bundling and alignment with thick (1−3 μm) bundles of collagen fibers aligned perpendicularly to the primary tumor front (Fig. 4a−e). Time-lapse, SHG imaging of the primary tumor fronts showed that control HEp3 cells actively invade along the aligned collagen bundles forming dominant cell protrusions that generally orient along the collagen fibers perpendicular to the primary tumor front (Fig. 4a−c, f and Supplementary Movie 5). In contrast, tumors comprised of HEp3 cells that express Kif3b targeting shRNAs failed to reorganize the

collagen fiber network at the primary tumor periphery. Collagen fibers at the Kif3b mutant tumor fronts appeared to be disorganized with an almost complete absence of collagen bundles. Rare invasive mutant KIF3B HEp3 cells formed short non-directional protrusions (Fig. 4a−f and Supplementary Fig. 4b, d; Supplementary Movie 5). Accordingly, when KIF3B mutant HEp3, HT1080, and MDA231 cancer cells were tested in a 3D collagen invasion assay, invasion of the KIF3B mutant cell lines was significantly reduced (Supplementary Fig. 6a, b). This suggests that our screening approach preferentially identified genes required for the coordination of directional in vivo cell migration and invasion.

**Inhibition of identified genes blocks spontaneous metastasis in vivo.** To test the hypothesis that genes required for in vivo cell motility and directional cell migration are also required for intravasation and metastasis, we evaluated each of the hit clones in a xenograft murine model of spontaneous metastasis to the lungs. To this end, we established subcutaneous HEp3 tumors in the flank of nude mice using scrambled shRNA control or KIF3Bsh/sh2, SRPK1sh/sh2, and NR2F1sh/sh2 expressing tumor cells. When the primary tumors reached 1.5 cm³, the lungs were examined for the presence of metastasis using whole-mount fluorescence stereomicroscopy and then quantified using human alu-specific q-PCR (Fig. 5a, b)[14]. In animals bearing shRNA scramble control HEp3 tumors (n = 23), significant metastasis to the lungs was detected by fluorescence imaging (Fig. 5a). In contrast, metastatic lesions were rarely observed in the lungs of animals bearing KIF3Bsh1/sh2, SRPK1sh1/sh2, and NR2F1sh1/sh2 tumors, and these were very small in size (Fig. 5b, d). To accurately quantify the burden of metastatic HEp3 cancer cells in the murine lungs, we extracted genomic DNA and performed human-specific alu q-PCR. The precise enumeration of metastatic cells in the lung was then determined by comparing these data to a standard curve generated from HEp3 cells[18]. The scramble shRNA control had an average of 2.4 million disseminated cancer cells per lung. In contrast, animals bearing KIF3Bsh1/sh2, SRPK1sh1/sh2, and NR2F1sh1/sh2 tumors had a dramatic inhibition of metastatic dissemination, with reductions in metastasis to the lungs of 99.55 and 99.67% respectively for KIF3Bsh1 and sh2, 99.98 and 99.66% respectively for SRPK1sh1 and sh2, and 99.71 and 99.81% respectively for NR2F1sh1 and sh2 (Fig. 5e). To confirm that this inhibitory effect is not cell type and shRNA gene knockdown technology limited we used CRISPR technology to knockout our top target, Kif3b, in the HT1080 fibrosarcoma cell line (Supplementary Fig. 7a). Subcutaneous injection of control HT1080 cells resulted in a robust spontaneous lung metastasis as confirmed by quantitative alu q-PCR, stereomicroscopic and IHC analysis (Supplementary Fig. 7a−c, e, g). In contrast, in Kif3b mutant cells metastasis was severely (~80%) inhibited forming smaller and less frequent lung metastatic lesions (Supplementary Fig. 7a-c,f,h). There was no significant difference in primary tumor growth rates between the control and hit shRNA clone tumors (Supplementary Fig. 7d and Supplementary Fig. 8a). These results confirm our hypothesis that genes required for in vivo cell motility and directional cell migration are also required for spontaneous metastasis, and that KIF3B, SRPK1, and NR2F1 represent promising therapeutic targets for metastasis.

Considering the possibility that the observed motility phenotypes could be specific to the highly metastatic human HEp3 epidermoid-carcinoma cell line, we evaluated the silencing of our top targets KIF3B, SRPK1, and NR2F1 using MATs in vitro cell migration model using HEp3, HT1080, MDA231, and PC3 human cancer cell lines. Silencing of Kif3b efficiently blocked in vitro cell

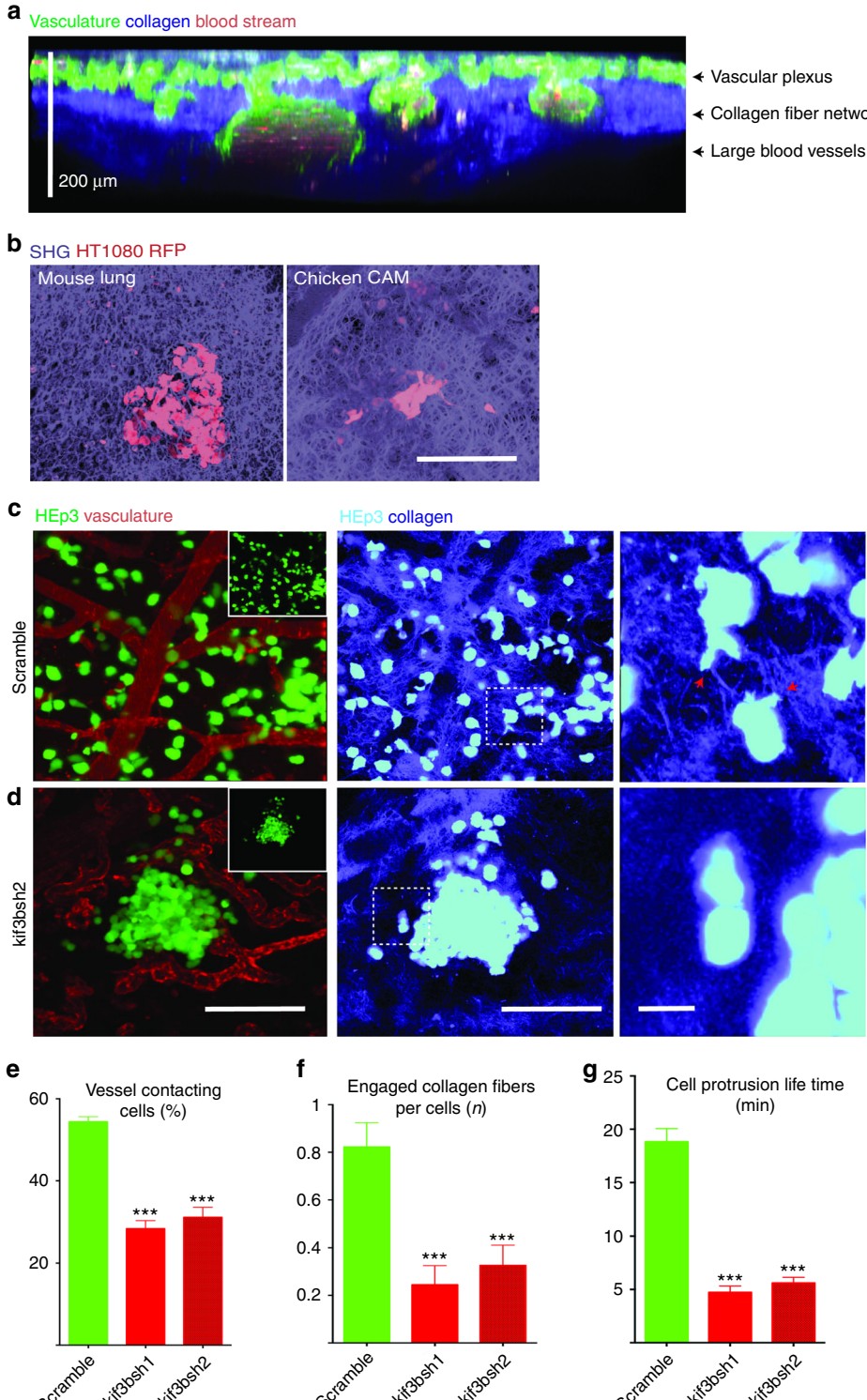

**Fig. 3** Kif3b is required for metastatic HEp3 cancer cell vasculotropism and invasion into the extracellular matrix in vivo. Multicolor two-photon intravital imaging was utilized to visualize the chicken CAM structure and cancer cell behavior within metastatic cancer cell colonies. **a** Chicken CAM structure, 15dpf. FITC-lectin (green) was used for visualization of the vasculature, SHG (blue) for imaging of collagen fiber network and auto-fluorescence (red) for imaging of the blood. **b** Representative images showing similarity in the collagen matrix structure between the mouse lung (left panel) and chicken CAM (right panel). Metastatic colonies that were formed by control (**c**, scramble shRNA) and Kif3b shRNA2-transduced HEp3 cells (**d**). Right panels in (**c**) and (**d**) show higher magnification of areas in middle (SHG) panels. Red arrows point to cancer cell protrusions that are in contact with collagen fibers. Note that control cells robustly interact with the vasculature and invade into the collagen matrix while shRNA2 Kif3b cells fail to do so. Insets show only tumor cell (GFP) channel. **e** Quantification of the fraction of cells in contact with blood vessels for HEp3 control and Kif3b knockdown cells. **f** Average number of cancer cell engaged collagen fibers for HEp3 control and shRNA2 Kif3b cells. **g** Average cell protrusion lifetime for control and shRNA2 Kif3b cells (see also Supplementary Movie 3). Scale bars = 200 μm (**a**, **b**); 200 μm (**c**, **d**, left and middle panels) or 20 μm (**c**, **d**, right panel)

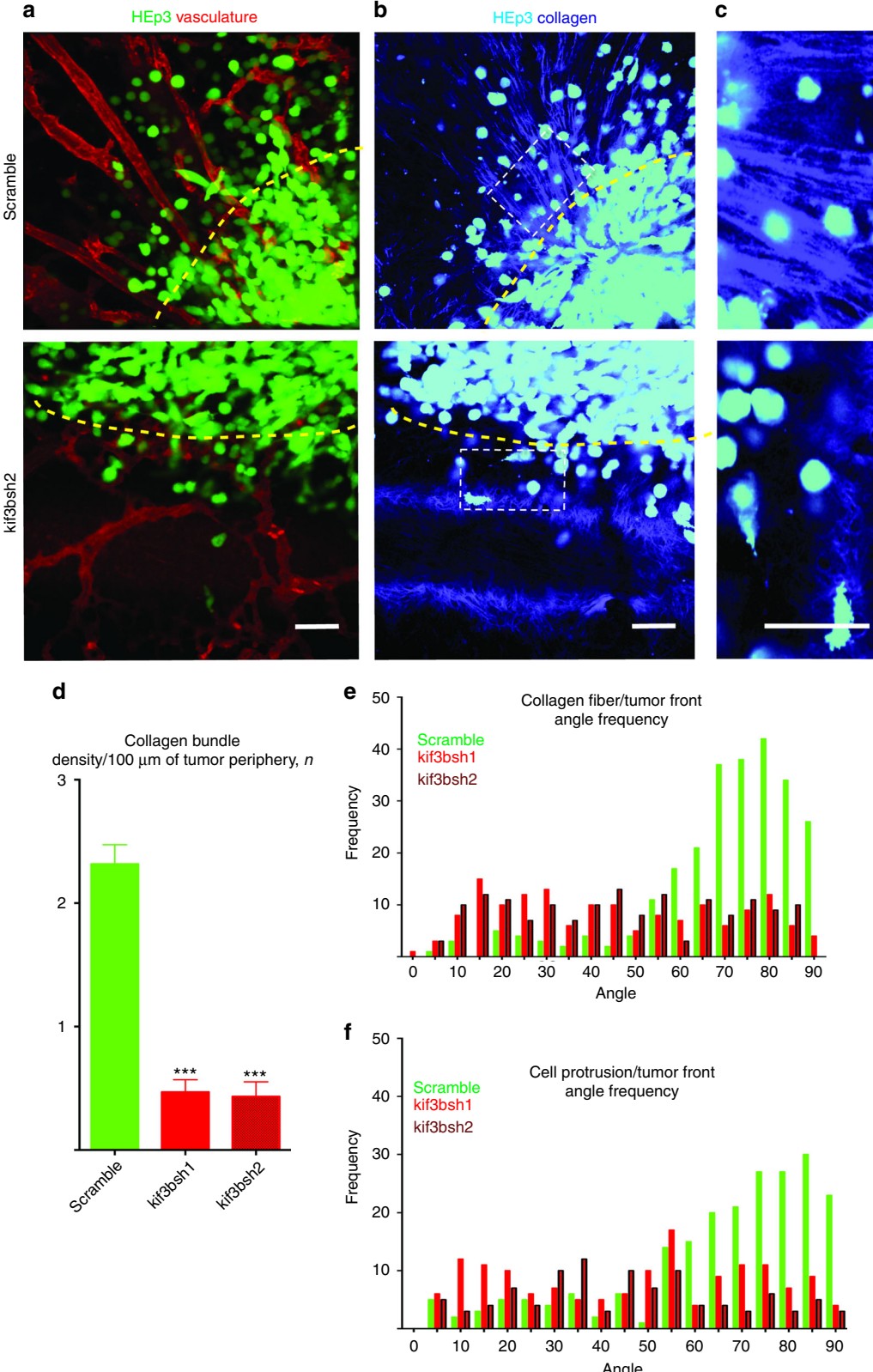

**Fig. 4** Kif3b is required for collagen fiber alignment at the tumor front in vivo. **a** Primary tumor fronts of HEp3 control (upper panel) and HEp3 shRNA2 Kif3b (lower panel) tumors as visualized using intravital confocal microscopy. **b** Collagen fiber organization (SHG) along the HEp3 control (upper panel) and shRNA2 HEp3 Kif3b (lower panel) tumor fronts, see also Supplementary Movie 5. **c** Higher magnification images of the collagen fiber network from within white dashed rectangles in (**b**). Yellow dashed lines delineate tumor borders. **d** Quantification of collagen bundle density at the invasive fronts of control and shRNAs1/2 Kif3b HEp3 tumors. **e** Quantification of collagen fiber alignment at the invasive fronts of control and shRNAs1/2 Kif3b HEp3 tumors. **f** Quantification of cancer cell protrusion orientation at the invasive fronts of control and shRNAs1/2 Kif3b HEp3 tumors. Scale bars = 100 μm

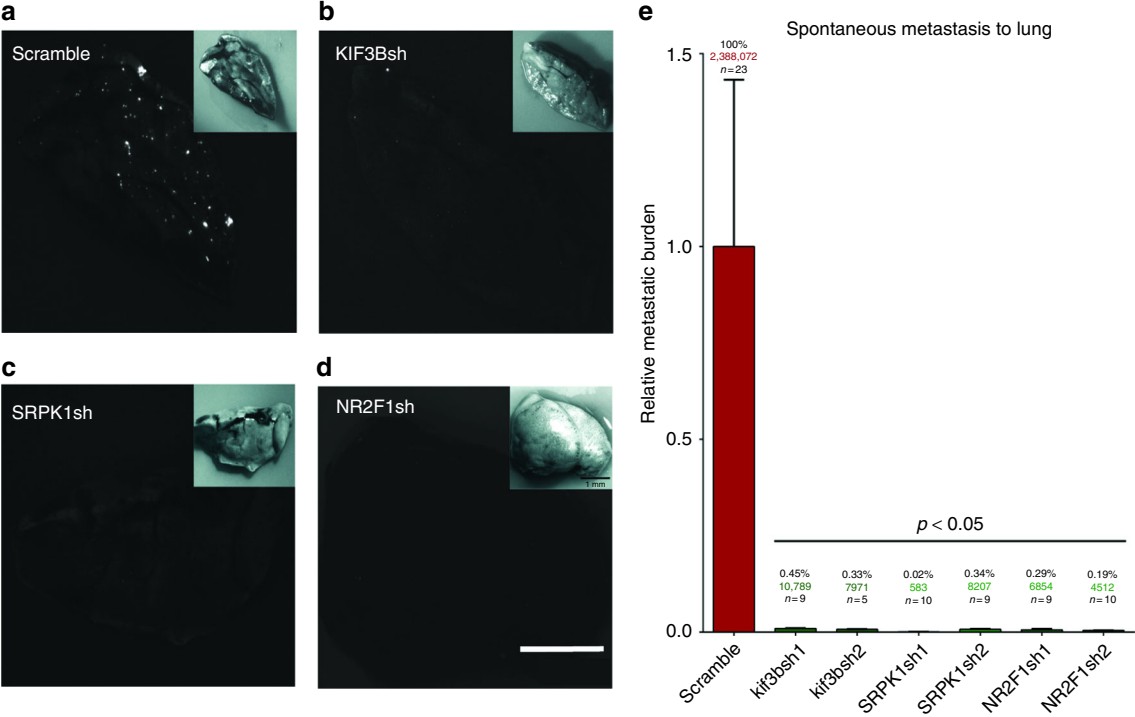

**Fig. 5** RNAi-mediated inhibition of screen-identified genes blocks spontaneous cancer cell metastasis in vivo. Fluorescence stereomicroscopic images of lungs from mice bearing subcutaneous tumors derived from HEp3 cancer cells transduced with **a** control (scramble) shRNA, **b** shRNA targeting KIF3B, **c** shRNA targeting SRPK1 or **d** shRNA targeting NR2F1. **e** Precise quantification of HEp3 cancer cells metastasized to lung as determined by human *alu* q-PCR. Data are expressed as relative metastatic burden in percentage, and as the total number of cancer cells detected (colored numbers) when estimated using a standard curve. Scale bar = 1 mm

migration in all of the cancer cell lines (Supplementary Fig. 8b). Interestingly, silencing SRPK1 significantly inhibited the in vitro motility of HEp3 and PC3 cells but had no effect on the motility of MDA-MB-231 (Supplementary Fig. 8c). Silencing of NR2F1 inhibited HEp3 migration in vitro but also had no effect on MDA-MB-231. No NR2F1 expression was detected in PC3 cells (Supplementary Fig. 8d). This supports the idea that screening in a functional in vivo microenvironment is important as these phenotypes are not necessarily recapitulated in 2D culture.

**KIF3B and SRPK1 are overexpressed in invasive prostate cancer.** Next we explored the potential relevance of these genes for human cancer progression and metastasis. To do this, we utilized human cancer gene expression databases (Oncomine) to evaluate associations between hit gene expression and cancer progression, metastasis or poor clinical outcomes[19]. Indeed, our analysis indicated that the top hit genes identified in our screen are significantly upregulated in the metastatic lesions of several solid cancer types including: melanoma (KIF3B, C14orf142, and NR2F1), prostate cancer (SRPK1 and KIF3B), head and neck cancer (KIF3B), lung cancer (SRPK1 and TMEM229B), ovarian cancer (NR2F1) and colon cancer (NR2F1) (Supplementary Fig. 9a). Moreover, a detailed survey of immunohistochemical staining of human cancers indicated that SRPK1, KIF3B, C14orf142, NR2F1, and TMEM229B have significantly increased expression in the invasive zone of the primary tumors of these cancers as delineated by a pathologist (Supplementary Fig. 9b–g). Quantitative analysis of an independent prostate cancer progression TMA cohort (University of Calgary) of 98 patients showed that both Kif3b and SRPK1 display significantly higher expression levels in prostate cancer epithelium compared to benign hyperplasia epithelium (Fig. 6a, b). Both Kif3b (Fig. 6c) and SRPK1 (Fig. 6d) were expressed at significantly higher levels in areas where the prostate epithelium is invading into the surrounding stroma, further correlating their overexpression in the process of prostate cancer invasion and metastasis.

## Discussion

We developed a powerful quantitative in vivo screening approach to discover therapeutic targets for metastasis based on productive cancer cell motility in a complex tumor microenvironment. Using a combination of the highly accessible ex ovo avian embryo model, fluorescence time-lapse intravital imaging and high throughput sequencing, the platform offers a rapid and quantitative means to detect and efficiently stratify clinically relevant phenotypes and identify potential therapeutic targets for metastasis. Indeed, all of the hits from this screen that have been selected for downstream analysis have shown to be effective targets for metastasis, as their inhibition results in a nearly total block of metastatic dissemination.

Several of the genes identified in the screen validate the approach as they have been previously linked with cancer cell invasion and/or migration. The identification of actin isoforms ACTB and ACTC1, for instance, is not surprising as they are central to the cell migration machinery and have been previously identified using in vitro RNAi screening approaches[20]. SRPK1 has been previously implicated in cancer progression and metastasis through its regulation of VEGF splicing[21–23], yet we did not observe an inhibition of primary tumor growth when SRPK1 expression was reduced. SRPK1 was also recently identified as a mediator of cell migration using an in vitro RNAi screening approach[21], which when taken together with our results, supports

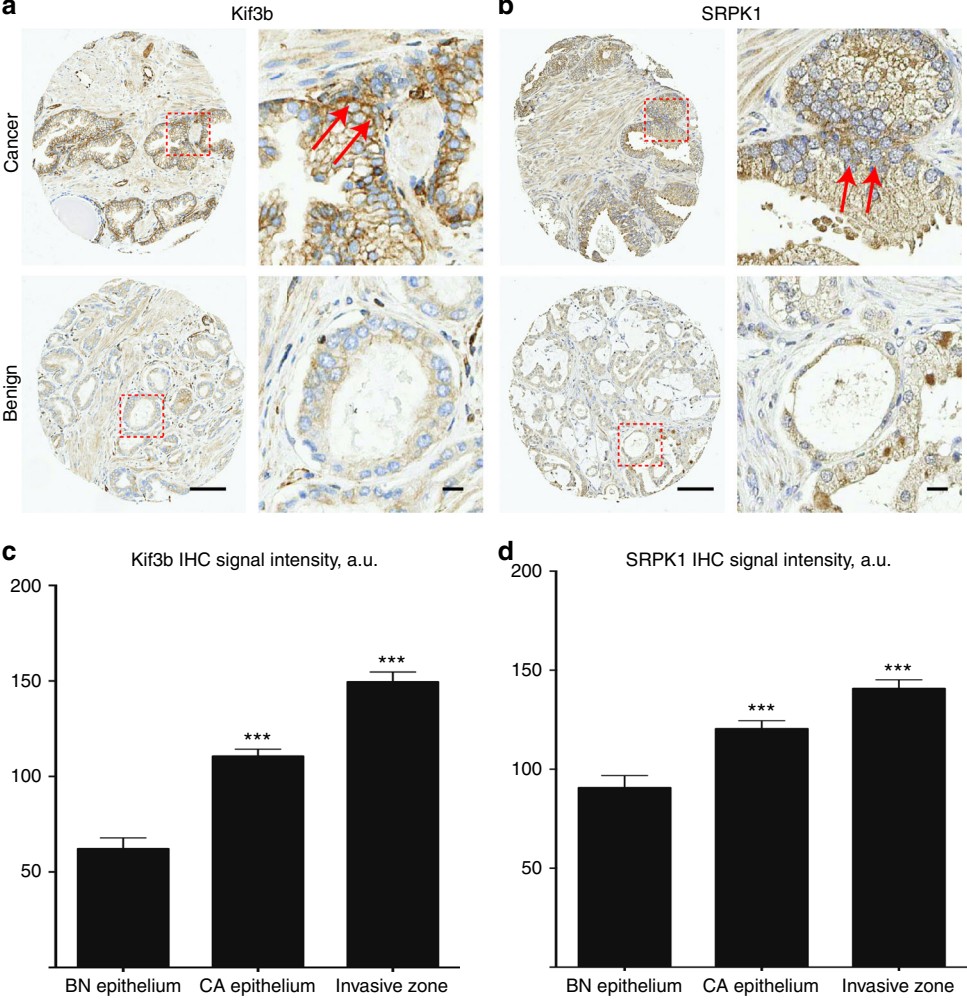

**Fig. 6** KIF3B and SRPK1 are overexpressed in invasive prostate cancer epithelium. The expression of KIF3B and SRPK1 in 98 prostate cancer patients was assessed by immunohistochemistry. Representative prostate tissue sections stained with Kif3b (**a**) and SRPK1 (**b**) antibodies in areas with prostate cancer (upper panels) and benign prostate hyperplasia (lower panels) samples. Right panels in (**a**) and (**b**) show enlarged areas from within the red dashed squares. Red arrows denote prostate cancer invasive zones. Quantification of KIF3B (**c**) and SRPK1 (**d**) expression within the epithelium in prostate benign hyperplasia, prostate cancer epithelium, and prostate cancer invasive zones. Scale bars = 100 μm (**a, b**, left panels) and 10 μm (**a, b**, right panels)

the idea that SRPK1 plays a more direct mechanistic role in cell migration and metastasis. It will be important, going forward, to identify the distinct but overlapping mechanisms responsible for productive cell migration in vivo compared to in vitro.

The screen identified several genes of known function that had not been directly implicated in cancer cell motility and metastasis. The availability of research tools and in some cases, small molecule inhibitors, for these targets highlights them as attractive candidates for further preclinical testing[22–25]. Identification of KIF3B as our top target is intriguing since this gene is part of the kinesin motor machinery responsible for transporting multiple therapeutically relevant molecules such as β-catenin and MT1-MMP[26–28]. Along with Rho kinase signaling, the precise kinesin-driven delivery of MT1-MMP to the cancer cell front is essential for collagen fiber matrix realignment and degradation[27,28]. This is consistent with our finding that targeting Kif3b expression efficiently inhibits these processes, blocking cancer cell metastasis. The screen also identified several genes and non-coding RNAs of unknown function, including C14orf142, TMEM229b, KB-1460A1.5, and KIAA0922, whose expression in human cancers would suggest a yet-to-be-described role in cancer progression and metastasis. The identification of putative regulatory pro-metastatic RNAs in particular warrants further investigation[29].

Overall, we found an extraordinarily tight concordance between a gene's in vivo cancer cell motility phenotype and its requirement for successful metastatic spread. This study demonstrates that our quantitative in vivo imaging-based screening approach is a powerful tool to identify potential therapeutic targets for metastasis, paving the way for the development of new therapies to block this deadliest aspect of human cancer.

## Methods

**Cells, antibodies, and reagents**. The human epidermoid-carcinoma cell line (HEp3), human breast cancer cell line (MDA-MB-231), and human prostate cancer cell line (PC3) were obtained from ATCC and maintained as described previously[13]. Human shRNAGIPZ microRNA-adapted shRNA lentiviral library was purchased from Open Biosystems. Control cells were labeled with non-silencing control viral stock (GFP expressing). For functional MOI calculation HEp3 cells ($10^5$ cells per single well in a 12-well plate) were infected with shRNAGIPZ library stocks in triplicates and the percentage of GFP-positive cells was quantified 48 h post infection using EVOS FL Cell Imaging System with six fields of view per MOI. An MOI = 0.2 was used for all the experiments. Individual, independent hairpin lentiviral vectors encoding shRNAs targeting SRPK1, NR2F1, KIF3B, TMEM229B, C14orf142, and ACTB (ready to use lentiviral particles) were purchased from Santa Cruz Biotechnology and used according to the manufacturer protocols. For Kif3b CRISPR gene knockout guide RNAs with high predicted on-target effects and low predicted off-target effects were chosen using the Benchling "Design CRISPR guides" tool (https://benchling.com/). Guide RNAs were cloned into CRISPR-Cas9 plasmid PX459 (pSpCas9(BB)-2A-Puro V2.0), a gift from Feng

Zhang (Addgene plasmid # 62988). Plasmids were expanded in competent *Escherichia coli* (Stbl3) and then sequenced. KIF3B CRISPR-Cas9 plasmids were transfected into HT1080 cells using Lipofectamine 2000. Forty-eight hours post-transfection, cells were selected using 1 μg/ml puromycin for 72 h. Surviving clones were expanded and pooled for downstream assays. CRISPR-Cas9 editing of KIF3B at the genomic level was confirmed using q-PCR.

Antibodies were from Sigma (SRPK1), Santa Cruz Biotechnology (KIF3B) and Cell Signaling (NR2F1) and Invitrogen (Tubulin). Human TMEM229B and C14orf142-specific qRT-PCR primer sets were purchased from Bio-Rad.

**Quantitative RT-PCR and western immunoblot analysis**. Total RNA was extracted from trypsinized cells with Trizol (Invitrogen). RNA (1 μg) was reverse transcribed into cDNA by random priming using the One-Step RT-PCR Kit (QIAGEN). Quantification of cDNA was done using SYBR Green Supermix for real-time q-PCR (Bio-Rad) with oligonucleotide sequences that specifically recognize GAPDH, TMEM229B or C14orf142. GAPDH was used as a control for total cDNA. Samples for western immunoblot analysis were prepared by lysing $1.2\times10^7$ cells of interest with 400 μl of lysis buffer (1% IGEPAL CA-630 in PBS with 1× protease inhibitor cocktail (Sigma)) on ice for 30 min and collection of supernatants from centrifugation of samples at 4 °C at $16,000\times g$ for 15 min. Twenty micrograms of protein was separated by 10% SDS-PAGE gels and transferred onto nitrocellulose membranes (Bio-Rad) using a Trans-blot Turbo Transfer system. Membranes were blocked in 50% blocking buffer (LI-COR) in PBS at room temperature for 1 h and incubated with primary antibody in 50% blocking buffer at 4 °C overnight. At room temperature, membranes were then washed three times with TBS-Tween (TBST; 0.1%), incubated with a loading control antibody for 1 h, washed three more times with TBST, incubated with secondary antibodies (ThermoFisher Scientific, Alexa Fluor® −680 or −750) for 1 h, and washed again three times with TBST. Proteins were then detected using the Odyssey® Classic Infrared Imaging System (LI-COR).

**Ex ovo chick embryo cancer xenograft model**. Fertilized White Leghorn chicken eggs were purchased from University of Alberta Poultry Research Centre and incubated in a humidified chamber at 38 °C. At day 4, embryos were removed from their shells using a Dremel tool with a cutting wheel and maintained under shell-less conditions, in a covered dish in a humidified air incubator at 38 °C and 60% humidity as previously described[3,12]. On day 10 of development, chicken embryos were injected intravenously with $2.5\times10^4$ HEp3 cells that were transduced with shRNAGIPZ library or control lentiviral particles. Metastatic colonies were allowed to grow for 6 days and individual compact metastatic colonies were excised using micro-dissection scissors. Excised colonies were expanded under selective (puromycin) conditions, cryopreserved at earliest possible passage (p1−3) and used for further analysis as necessary. For time-lapse imaging of metastatic colonies, sterilized coverslips were applied on top of the CAM that contains metastatic colonies 24 h post tumor cell application. For imaging studies involving modeling of primary tumor fronts tumors, day 10 chicken embryos had $2\times10^5$ HEp3 cells in serum-free media injected directly in between CAM ectoderm and endoderm layers. Sterilized coverslips were applied on top of the tumor 24 h post tumor cell application and invasive fronts were imaged 5 days post tumor cell inoculation.

**3D collagen invasion assay**. HEp3, HT1080, or MDA231 human cancer cells ($10^5$ cells/well) stably labeled with GFP were cultured on the top of 3D collagen gel matrixes (2.5 mg/ml, ~200 μm thickness, eight-well Lab-Tek chambered slides) for 36 (HT1080, HEp3) or 48 h (MDA231) as described elsewhere[28]. 36–48 h later random 3D image fields were obtained using Nikon A1r MP two-photon microscope. Number of cells that invaded ≥100 μm from the gel surface was quantified manually. At least ten independent fields from two independent wells were analyzed for each condition.

**Image acquisition and analysis**. Real-time imaging of cancer cell invasion was performed by acquiring four-dimensional image series of single cancer cells within the CAM. A 50−60 μm image stack was acquired every 15 min in 1–3 μm step size increments for 2–8 h. A Zeiss upright microscope (Carl Zeiss Inc.) fitted with a temperature-regulated enclosure (Plastics Inc.), an XY stage controller (Ludl Inc.), a 405/491/561/646/750 nm diode laser switcher (Quorum Technologies Inc.), a Hamamatsu 512 × 512 EMCCD camera (Hamamatsu Inc.) and full range of Zeiss microscope objectives were used for image acquisition. Velocity Acquisition software (Improvision Inc.) was used to control the microscope and acquire all images. Image analysis and processing was performed using Velocity and ImageJ (NIH). Nikon A1r MP two-photon microscope equipped with ×25 WI objective was used for multicolor imaging of cancer cells and chicken embryo CAM vasculature. 860 nm (Spectra Physics Insight DS + laser, NDD acquisition mode) wavelength was used for SHG chicken CAM collagen imaging. Image drift was corrected using the ImageJ Stack_Reg plugin (Biomedical Imaging Group, http://bigwww.epfl.ch/) prior to tracking analysis. Time 0 was defined as the time of the first image capture. At least 20 individual cells were tracked for control cells and each knockdown clone (Fig. 2c–f) using the built-in Velocity Object tracking module. Track velocity was calculated as average speed of the track. Track displacement rate (productivity) was

calculated using the built-in Velocity module as total track displacement (straight line distance from the first track position to the last) divided by track time. For quantification of invasive cells per field, snapshots of invasive fronts for least ten independent tumors (×10 magnification) were analyzed for each condition (Fig. 2g). For cell protrusion quantification, invasive tumor fronts were imaged at ×20 magnification and at least 25 individual cells were analyzed for each condition (Fig. 2h). Invasive cells per field and cell protrusion numbers were quantified manually using ImageJ software. Collagen fiber and cell protrusion angle was quantified using the ImageJ built-in angle measurement module ("acute", sharper angle was used for each measurement). At least 150 collagen fibers or 100 cell protrusions (3–10 independent tumors for each condition) were analyzed. Results were plotted as angle frequency distribution, bin = 10, built-in Prism analysis module.

**Colony compactness quantification**. We used MATLAB to derive a composite colony compactness index (C.I.) based on three different colony attributes: cell distribution within a colony, colony area, and colony density. Images of individual colonies were converted to grayscale (rgb2gray function) and had background signal subtracted. Uneven illumination was corrected using top-hat filtering (imtophat function) using a 10-pixel radius disk-shaped structural element. The colony centroids were determined (regionprops function) and images were translated so that colony centroids were in the center of images. Pixel-intensity line plots along each image row were determined (improfile function) and line plots for all rows were averaged. For each colony image, row average line plots were calculated for 18 different images that were rotated 20° from each other (360° total rotation) and row average line plots over all 18 rotated images were averaged, creating a single line plot. This line plot was fitted to a Gaussian distribution and the standard deviation of the distribution was the measure of cell distribution within colonies. Colony area was determined using the "bwconvhull" function on binary images of colonies with only cancer cells visible. Colony density was determined as the fraction of pixels within the colony convex hull area that contained cancer cells.

$$Z(\text{colony attribute}) = \frac{\text{Clone attribute mean} - \text{Control attribute mean}}{\text{Control attribute standard deviation}}$$

Colony compactness index (C.I.) values were determined which represents the relative level of compactness for each hit / clone. To ensure higher C.I. values for more compact colonies, the Z(cell distribution – Linear Index) and Z(colony area – Area Index) were subtracted from Z(colony density – Density Index) since Z(colony density) was positive in compact colonies while Z(cell distribution) and Z(colony area) were negative in compact colonies.

$$\text{Compactness Index(C.I.)} = Z(\text{colony density}) - Z(\text{cell distribution}) - Z(\text{colony area}).$$

Five to ten embryos were re-injected for each clone and CAM/metastatic colonies images were randomly acquired 5 days post injection using Zeiss Lumar V12 stereoscope equipped with ×1.5 objective (see Supplementary Fig. 4 for example). Single metastatic colony images were randomly cut out digitally and used for C.I. quantification. Ten randomly chosen metastatic colony images were analyzed for each of the screen hits. Z-scores were determined for all three colony attributes. For quantification of HT1080 and MDA231 colony compactness ×25 images (Nikon A1r microscope) were used. Similar to HEp3 cells at least ten randomly selected metastatic colony images were used. The C.I. quantification module exists as a stand-alone package and is available upon request along with sample metastatic colony images.

**In vitro MAts migration assay**. In vitro MAts migration assay was performed as previously described[30]. Briefly, control and mutant cancer cells (HEp3, MDA-MB-231, and PC3) were seeded in a 12-well culture plates with pre-attached MAts 12 h before the assay to achieve sub-confluency (70–80%) at the day of experiment. No significant effect on cancer cell proliferation was detected in shSRPK1, shKIF3B, shNR2F1 or scramble shRNA expressing cells (data not shown). At the day of the assay MAts were removed and cells were allowed to migrate for 6-12 h depending on the cell line tested. All assays were done in triplicate and images were acquired using an EVOS FL Cell Imaging System, ten fields of view per condition. Data analysis to determine percentage of total pixels that were not covered (percentage open area) was done using TScratch software version 1.0 for Windows with MATLAB Compiler Runtime (Swiss Federal Institute of Technology in Zürich, Zürich, Switzerland).

**Mouse spontaneous metastasis model**. The ability of tumors derived from KIF3Bsh/sh2, SRPK1sh/sh2, and NR2F1sh/sh2 HEp3 cancer cells to spontaneously metastasize from primary tumors to the lung was compared to control using a xenograft athymic (nu/nu) mouse model. $1\times10^6$ HEp3-GFP or HT1080-RFP cells were implanted subcutaneously in the rear flank of week 21 female athymic (nu/nu, HEp3 cells) or SCID (HT1080 cells) mice and the resulting tumors were monitored until they reached a volume of 1500 mm³. At this point, the lungs were harvested

and visualized with a Zeiss Lumar V12 fluorescence stereomicroscope with bright field and GFP filters. To precisely quantify the burden of human HEp3 cancer cells in the mouse lungs, q-PCR was performed with primers specific for human *Alu* repeats and mouse GAPDH. Quantitative PCR using *Alu*-specific primers was performed using the SYBR green amplification from the Extract-N-Amp kit as previously described[14]. Quantitative PCR using mouse GAPDH primers was used to normalize for tissue input. A standard curve was generated using a serial dilution of HEp3 cells at known concentrations and compared to the in vivo results to estimate absolute cancer cell numbers. All animals were housed, maintained and treated by procedures approved by University of Alberta Institutional Animal Care and Use Committees (IACUC).

**Oncomine and Protein Atlas analysis**. An analysis of stained tissue sections obtained from the Human Protein Atlas (www.proteinatlas.org) was conducted by a certified pathologist (B.A.). Only the samples with medium to high IHC scoring intensity in either normal prostate or cancer tissue were included in invasive tumor front identification. Detailed information regarding the Protein Atlas sample quality assurance can be found at http://www.proteinatlas.org/qc.php.

A meta-analysis of individual genes across publicly accessible prostate cancer expression databases was performed using the meta-analysis tool within Oncomine (https://www.oncomine.com), the steps of which have been previously described in detail[19]. Briefly, microarray data from individual studies were normalized, and differential expression between pre-specified classes was determined by Student's *t* test. Genes were ranked within each dataset by the *p* value, and enrichment (positive or negative) across studies was determined by comparing observed rankings against a random distribution. Studies selected had a patient cohort >20 and allowed differential expression analysis within Oncomine. Patient annotated gene expression heat maps were generated using Oncomine.

**Prostate cancer progression TMA analysis**. IHC was performed on a progression TMA (University of Calgary) from 98 patients, for a total of 320 cores. Tissue samples targeted included benign ($n = 80$) and localized PCa ($n = 162$). Diagnosis of TMA cores was confirmed by the study pathologists (T.A.B.). IHC stain was performed using auto-stainer of Dako Omnis (Agilent, Santa Clara, CA, USA) for KIF3B or Leica BondMax (Leica Biosystems Inc. Buffalo Grove, IL, USA) for SRPK1. Anti KIF3B and SRPK1 antibodies were purchased from Sigma (St. Louis, MO, USA).

Four micron FFPE sections were pretreated with heat-induced epitope retrieval procedure that was carried out on the auto-stainer as routine IHC procedure. Citrate pH 6.0 epitope retrieval buffer was for KIF3B and SRPK1. Antibody was diluted to 1/100 to 1/400-fold using Dako antibody diluent. Primary antibody incubation was carried out for 30 min on Omnis or 15 min on Bond, while secondary incubation was carried out for 30 min on Omnis or 8 min on Bond. FLEX DAB + Substrate Chromogen system for Omnis or Bond Polymer Refine Detection kit for BondMax was used as post incubation detection reagent.

To quantify the protein expression, images were converted into 8-bit grayscale, inverted and average intensity within region of interest was quantified using ImageJ software. Antibodies were from Abcam (SRPK1) and Sigma (Kif3b).

**Statistics**. All experiments were repeated at least three times. Data are presented as the mean ± s.e.m. Unless otherwise indicated, data were tested for significance using one-way ANOVA with Fisher's LSD test. The level of significance was set at $p < 0.05$.

**Data availability**. Authors can confirm that all relevant data are included in the paper and/or its Supplementary Information files.

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

## Acknowledgements

We thank Katia Carmine-Simmen, Amber Ablack, and Douglas Brown for their valuable technical support, Desmond Pink for key assistance with ex ovo embryo culture and imaging. This study was supported by the Canadian Cancer Society Research Institute Grant #700537 to J.D.L., who holds the Frank and Carla Sojonky Chair in Prostate Cancer Research supported by the Alberta Cancer Foundation.

## Author contributions

K.S., L.W., and J.D.L. designed the experiments, interpreted the results, and wrote the manuscript. K.S and L.W. performed majority of the experiments in the manuscript. R.J.P. designed and performed the C.I. image analysis. D.J.B., S.R., A.E.R., and F.W. contributed to the intravital screening experiments. J.J. and G.K.-S.W. conducted high throughput sequencing. B.A. assisted with Protein Atlas and Oncomine data analysis. T.A.B. performed and analyzed the prostate cancer TMA staining. E.W. performed western blotting, qPCR experiments and contributed to the intravital screening

experiments. D.L.S. contributed to the animal experiments. A.Z. assisted with data analysis and results interpretation.

## Additional information

**Competing interests:** The authors declare no competing interests.

