## [Peer Review File · Nature Communications]

Reviewers' comments:

Reviewer #1 (Remarks to the Author):

The manuscript describes a screening platform for quantitative *in vivo* identification of novel genes involved in cellular motility. The established platform provides a surrogate for classic Boyden-chamber based screens. Although the approach holds potential for large-scale projects, the model system (avian embryos) largely excludes tumor microenvironmental factors. Similarly, the systemic injection of highly invasive human cancer cells does not allow studying intravasation in the current approach. Overall, the manuscript could benefit and evolve by considering the following comments.

1. The authors highlight intravasation as a major rate-limiting step in the metastatic cascade. The quantitative *in vivo* screening platform established here employs a systemic intravenous injection of tumor cells in *ex ovo* avian embryos. However, this artificial starting point not only skips crucial intravasation steps, but also floods the system with a large number of tumor cells, thereby failing to truthfully recapitulate *in vivo* metastasis progression.

2. The authors claim that they could screen therapeutic targets for metastasis based on cell motility in a complex tumor microenvironment. However, it is not quantified to what extent human tumor cells will interact with avian embryonic cells. Further, this reviewer is not convinced whether the developing avian embryo harbors similar matrix properties as found in the metastatic niche. Lastly, the sheer lack of immune system questions the predictive utility of this approach.

3. The authors illustrate reduced metastatic burden in the lungs of tumor-bearing mice. A detailed analysis of primary tumor, including the tumor growth curve as well as analysis of primary tumor cell velocity, displacement rate and protrusions per cell, would provide a mechanistic understanding for identified genes of interest. Likewise, the authors should compare the numbers of circulating tumor cells in the blood of tumor bearing mice, as to differentiate between intravasation and colonization

4. The quantification of spontaneous lung metastatic foci needs to be included and the authors might include the analysis of tumor cell track velocity, displacement rates and protrusions per cell at metastatic foci. Such a comprehensive analysis would strengthen the current manuscript.

5. Generally speaking, human cancers exhibit a wide inter- and intra-tumoral diversity in gene expression. Thus, to explore the potential relevance of identified target genes, the authors must undertake a quantitative analysis of immunohistochemical staining in a cohort of patient samples.

6. The authors utilized only HEp3 cells for their experiments. However, considering the extremely invasive behavior of these cells, it would be advisable to use a second model (probably a less invasive model) to validate the findings *in vivo*, especially in the context of intravasation and invasive front at primary tumor site. It would be even better to use a murine-derived cell line as it would ensure appropriate interactions with the tumor microenvironment.

Reviewer #2 (Remarks to the Author):

This manuscript by Stoletov et al use a sophisticated genome-wide screen to identify novel regulators of metastasis *in vivo*. They go on to validate that three of the targets regulate metastasis in mice and further evaluate their expression in human cancers. The screen is elegant and powerful to identify regulators of metastasis *in vivo*, especially coupled with the *in vivo* imaging and measurement of cell dynamics. The validation in mice and human samples is important, but is less well developed. Additional work is needed to further validate the targets to support the robustness of the screen.

Specific comments:

1. Page 6, line 7 "A minimum of 10 individual colonies for each clone were analyzed". What does this mean? What are individual colonies in each clone?
2. It's not clear to me what the C.I. represents or how the calculation was arrived at ($C.I. = Z(\text{density index}) - Z(\text{linear index}) - Z(\text{area index})$). Why are Z scores for linear and area indices subtracted from the Z score for density index?
3. While there is a difference between the velocity in the control and shRNA knockdown samples, this difference is not noted in the primary tumor. From the absolute velocity, it appears that in the metastatic sites, the average is about 3.5 $\mu\text{m}/\text{sec}$ and the knockdowns are about 1 $\mu\text{m}/\text{sec}$. In the primary sites, all of them are $\sim 1\mu\text{m}/\text{sec}$. So, the difference appears to be only in the control sample, which migrates faster at the metastatic sites, whereas the knockdowns migrate the same speed in primary and metastatic sites. How can this be explained?
4. In some experiments, knockdown in MDA-MB-231 cells does not produce an effect on cell migration. The authors claim this indicates a difference between 2D and in vivo migration. To support this claim, in vivo migration/metastasis is necessary with these cells. Furthermore, the knockdown in these cells seems relatively inefficient, and the lack of effect could simply be due to residual protein.
5. The phenotypes in the ex ovo avian embryo and mouse metastatic models are somewhat different. In the avian model, the cells seed, but fail to spread, whereas in mice, they appear not to efficiently seed the primary tumor. What could cause this discrepancy? Are there changes in dissemination (i.e. circulating cells?) or invasion at the primary tumor in mice?
6. Why were only some genes evaluated in some human cancers? Are the ones shown in Figure 4 the only ones evaluated, or were the others examined but are not differentially expressed in primary and metastatic sites? Are the genes evaluated differentially expressed in primary and metastases in the avian or mouse models? Also, are any of the samples paired primary/metastasis?
7. There is no information on the human tissue sections in the methods. How many were analyzed? Staining procedures? Antibody validation? etc. Also, increased expression at the invasive edge is not apparent from the images, and quantification would strengthen this argument.
8. In the methods it states that at least 20 cells were tracked for each control and knockdown clone, but how many clones were evaluated for each? How were the 20 cells for each selected? There needs to be more description of how the velocity and displacement (i.e. productivity of cell migration) were calculated. Why are velocities reported as $\mu\text{m}^2/\text{sec}$? Why is the μm unit squared? Mean squared displacement would also be a valuable measurement to compare movement rates. Was displacement calculated over the entire track?
9. "Productivity of cell migration" is used throughout the text, but is never well defined, except in the figure legends. A more descriptive term or clearer definition is needed in the text.
10. In a few instances the authors claim that their screening approach demonstrates a powerful tool to identify novel therapeutic targets. This is somewhat overstated, since there is no validation of whether the genes identified are actually good therapeutic targets.
11. A description of panel f is missing from the figure legend for supplementary figure 3.
12. There is a typo in the label in figure 2a, bottom panel.

Response to Reviewer #1

1. *“The manuscript describes a screening platform for quantitative in vivo identification of novel genes involved in cellular motility. The established platform provides a surrogate for classic Boyden-chamber based screens. Although the approach holds potential for large-scale projects, the model system (avian embryos) largely excludes tumor microenvironmental factors.”*

Author response: While we agree that the avian embryo model is well-suited to large-scale projects as we've described here, we respectfully disagree with the assertion that the chorioallantoic membrane “provides a surrogate for classic Boyden-chamber based screens”. There is a considerable established body of literature around the CAM as an *in vivo* model of human cancer going back more than 30 years (a few notable examples include Eur J Cancer. 1969 Jun;5(3):287-95; Cell. 1996 May 31;85(5):683-93; J Cell Biol. 1998 Mar 9;140(5):1255-63; Blood. 2003 Jul 1;102(1):184-91; Nat Med. 2006 Mar;12(3):354-60; Proc Natl Acad Sci U S A. 2006 Jan 31;103(5):1475-9; Cancer Cell 2008, 13:221-234; Oncogene. 2013 Jan 17;32(3):363-74; Neoplasia. 2014 Oct 23;16(10):771-88; Cancer Res. 2014 Jan 1;74(1):173-87; Cell Rep. 2017 Apr 18;19(3):601-616). The complexity of the avian CAM microenvironment is roughly equivalent to that of immunocompromised mouse or rat lung yet it is accessible for intravital imaging without surgical intervention or window implantation. In the **revised Figure 3b**, we compare the tissue structure of mouse lung and CAM by multi-photon confocal imaging with second harmonic generation to visualize the collagen network and vasculature (also see images below).

HT1080 RFP Collagen (SHG)

In addition, we have completed a number of additional experiments to elucidate the impact of screen hits on the interaction of cancer cells with the microenvironment. In the revised manuscript, we demonstrate that reducing Kif3b expression in HEP3 cancer cells impedes their ability to interact with blood vessels and the local collagen network *in vivo* (**revised Figs. 3 and 4**). We further show that these findings are consistent in other cancer human cell lines such as MDA231 and HT1080 (**revised Supplementary Figs. 4-6**).

2. *“The authors highlight intravasation as a major rate-limiting step in the metastatic cascade. The quantitative *in vivo* screening platform established here employs a systemic intravenous injection of tumor cells in *ex ovo* avian embryos. However, this artificial starting point not only skips crucial intravasation steps, but also floods the system with a large number of tumor cells, thereby failing to truthfully recapitulate *in vivo* metastasis progression.”*

Author response: While the reviewer is correct that our screen doesn't assess intravasation directly, we believe the reviewer misunderstood our rationale which is explained on page 1 (paragraph 2), where we state that we have previously shown (Cancer Cell 2008, 13:221-234 and Cancer Research 2014, 74:173-87) that **intravasation rates are directly related to *in vivo* motility**. Therefore, “we sought to develop an *in vivo* approach to feasibly screen for genes required for motility, and thus intravasation and metastasis”. The goal of this paragraph is to highlight the fact that tumor cell motility is necessary for cancer cell metastasis, including intravasation. Moreover, even though cancer cell motility was a primary readout in our screen, we found that genes required for *in vivo* motility are required for successful metastasis from a primary tumor (**Fig. 5 and Supplementary Fig. 6**). We therefore left this paragraph unchanged.

3. *“The authors claim that they could screen therapeutic targets for metastasis based on cell motility in a complex tumor microenvironment. However, it is not quantified to what extent human tumor cells will interact with avian embryonic cells. Further, this reviewer is not convinced whether the developing avian embryo harbors similar matrix properties as found in the metastatic niche. Lastly, the sheer lack of immune system questions the predictive utility of this approach.”*

Author response: To address this point we now include additional quantitative data showing that blocking the expression of our top screen-identified target, Kif3b, inhibits the interaction of tumor cells with the avian CAM vasculature and extracellular matrix (**Figs. 3 and 4, Supplementary Movies 3-5**). We have shown that, similar to what is seen in human cancer patients (**refs. 16-17**), blocking of Kif3b expression prevents tumor cell interaction with the collagen matrix and tumor cell-induced collagen fiber alignment (**revised Figs 3-4, Supplementary Movies 3-5**). It is clear from this high-resolution intravital imaging data that the chicken CAM microenvironment is similar to that in mice and humans. We and others have published extensively on the interaction of human tumor cells with the chicken embryo immune complement (eg. *Blood*. 2006 107:317-27), which again is similar to that of an immunocompromised mouse.

4. *“The authors illustrate reduced metastatic burden in the lungs of tumor-bearing mice. A detailed analysis of primary tumor, including the tumor growth curve as well as analysis of primary tumor cell velocity, displacement rate and protrusions per cell, would provide a mechanistic understanding for identified genes of interest. Likewise, the authors should compare the numbers of circulating tumor cells in the blood of tumor bearing mice, as to differentiate between intravasation and colonization.”*

Author response: To address this point, we have added primary tumor growth curves for all the mouse tumor experiments (**revised Supplementary Figs. 6 and 7**). For HT1080 cell line, we now show that lung metastatic lesions that are formed by a Kif3b CRISPR knockout mutant are less frequent and smaller in volume than those of the control variant (**revised Supplementary Fig. 6**). To complement the already significant data describing tumor cell behavior at the primary tumor front (cell velocity, displacement rate and protrusion number in Figure 2, we have added additional quantitative data that describes tumor cell protrusion behavior (protrusion life time and directionality, cancer cell interaction with vasculature and ECM (**new Figs. 3, 4 and Supplementary Movies 3-5**).

One of the primary goals of this work was to develop a new intravital screening platform that does not require surgical intervention or highly specialized imaging equipment, as is required in the mouse. We clearly demonstrate that the genes discovered in the avian embryo cancer cell motility screen are required for spontaneous metastasis in mice (**Fig. 5 and new Supplementary Fig. 6**) and are consistent with human cancer patient data (**new Fig. 6 and Supplementary Fig. 8**). Therefore, we think that it is completely unnecessary to repeat all of the intravital imaging experiments in mouse metastasis models, where specialized surgical intervention and specialized equipment is required.

5. *“Generally speaking, human cancers exhibit a wide inter- and intra-tumoral diversity in gene expression. Thus, to explore the potential relevance of identified target genes, the authors must undertake a quantitative analysis of immunohistochemical staining in a cohort of patient samples.”*

Author response: The authors agree with this insight and in the original manuscript version we showed that two of our top screen hits Kif3b and SRPK1 overexpressed in human prostate cancer metastatic lesions and invasive zones using publically available immunohistochemical data (**Supplementary Fig. 8a,c**). To address the reviewer’s request we have performed additional new experiments to stain a prostate cancer progression tissue microarray (TMA) to examine the expression Kif3b and SRPK1. We show in the **new Fig. 6** that Kif3b and SRPK1 are overexpressed in human prostate cancer epithelium and the invasive zone of human prostate cancer.

6. *“The authors utilized only HEp3 cells for their experiments. However, considering the extremely invasive behavior of these cells, it would be advisable to use a second model (probably a less invasive model) to validate the findings in vivo, especially in the context of intravasation and*

invasive front at primary tumor site. It would be even better to use a murine-derived cell line as it would ensure appropriate interactions with the tumor microenvironment.”

Author response: To address this request, we have performed a number of additional *in vitro* and *in vivo* experiments. In the original version of manuscript we used HEp3 carcinoma, HT1080 fibrosarcoma, MDA231 breast cancer and PC-3 prostate cancer for our *in vitro* experiments and HEp3 for our *in vivo* experiments. To address the reviewer’s concern we have significantly expanded these studies to demonstrate that downregulation of Kif3b gene expression in HT1080 and MDA231 cancer cell lines inhibits the formation of invasive metastatic lesions *in vivo* (**new Supplementary Fig. 4**). Additionally, we have performed new experiments to demonstrate that HT1080, HEp3 and MDA231 cancer cells deficient in Kif3b expression fail to invade into 3D collagen matrixes (**new Supplementary Fig. 5**). Furthermore, we now demonstrate that knockdown of Kif3b gene expression in HT1080 blocks spontaneous metastasis in mice (**new Supplementary Fig. 6**).

Response to Reviewer #2

1. Page 6, line 7 “A minimum of 10 individual colonies for each clone were analyzed”. What does this mean? What are individual colonies in each clone?

Author response: To address this question we have updated our Materials and Methods section (see “Colony compactness quantification” section) as well as the legends to **Supplementary Fig. 2** to provide more detail on C.I. quantification and give examples of how colonies were chosen.

2. “It’s not clear to me what the C.I. represents or how the calculation was arrived at (C.I. = $Z(\text{density index}) - Z(\text{linear index}) - Z(\text{area index})$). Why are Z scores for linear and area indices subtracted from the Z score for density index?”

Author response: We agree with the reviewer, this could have been explained more clearly. We have updated the Materials and Methods section to describe the Colony compactness quantification in more detail. Briefly, to ensure higher C.I. values for more compact colonies, the Z(Linear Index) and Z(Area Index) were subtracted from Z(Density Index) since Z(Density Index) was positive in compact colonies while Z(cell distribution) and Z(colony area) were negative in compact colonies. The relationship between the Index scores and compactness can be seen in **Supplementary Fig. 2b-d**.

3. “While there is a difference between the velocity in the control and shRNA knockdown samples, this difference is not noted in the primary tumor. From the absolute velocity, it appears that in the metastatic sites, the average is about 3.5 $\mu\text{m}/\text{sec}$ and the knockdowns are about 1 $\mu\text{m}/\text{sec}$. In the primary sites, all of them are $\sim 1 \mu\text{m}/\text{sec}$. So, the difference appears to be only in the control sample, which migrates faster at the metastatic sites, whereas the knockdowns migrate the same speed in primary and metastatic sites. How can this be explained?”

Author response: This is an interesting observation! We noticed this phenomenon while acquiring time-lapse data. We found, surprisingly, that some Nr2f1 knockdown cells migrated more rapidly than controls at the primary tumor periphery yet were less directional. We believe that a possible explanation for this is related to the cancer cell’s ability to both model and interact with collagen matrix. In the revised manuscript we have conducted a number of intravital imaging experiments to visualize the collagen network using second harmonic generation (SHG) imaging. In striking contrast to tumors comprised of control cells, where the collagen network is dense and highly aligned at the tumor front, we find that in tumors comprised of Nr2f1-sh2 cells, for example, that the collagen network is less

dense, more disorganized and the mutant cells do not interact well with collagen matrix/fibers. In tumors comprised of mutant cells, the cancer cells are generally more free to move at the tumor front, but do not track along the collagen fibers. We think this effect is more pronounced in the primary tumor due to the large numbers of cancer cells compared to those in metastatic colonies. Metastatic colonies are typically lodged in the pre-existing dense collagen network of the CAM, and the mutant cells are deficient in their ability to remodel and/or engage the existing collagen network. New *in vivo* imaging data supporting this are seen in the **new Figs. 3 and 4**. We are currently following up this study and doing initial experiments visualizing the extravasation of control or Kif3b mutant cancer cells. However these experiments are outside the scope of this manuscript and therefore are not included.

4. “In some experiments, knockdown in MDA-MB-231 cells does not produce an effect on cell migration. The authors claim this indicates a difference between 2D and *in vivo* migration. To support this claim, *in vivo* migration/metastasis is necessary with these cells. Furthermore, the knockdown in these cells seems relatively inefficient, and the lack of effect could simply be due to residual protein.”

Author response: In our models MDA231 cells are less migratory and invasive than HEP3 and HT1080. They form significantly smaller metastatic colonies and display decreased collagen invasion. To answer the comments of both reviewers we have performed additional *in vivo* (**Supplementary Fig. 4**) and *in vitro* experiments (**Supplementary Fig. 5**) using MDA231, HT1080 and HEP3 cells. In all cases, inhibiting the expression of our top screen hit, Kif3b, led to significantly decreased invasion.

5. “The phenotypes in the *ex ovo* avian embryo and mouse metastatic models are somewhat different. In the avian model, the cells seed, but fail to spread, whereas in mice, they appear not to efficiently seed the primary tumor. What could cause this discrepancy? Are there changes in dissemination (i.e. circulating cells?) or invasion at the primary tumor in mice?”

Author response: These apparent differences can likely be explained (1) because the experimental approaches in each model are slightly different and (2) the way we presented the data. In the avian model primary tumor invasion (spontaneous metastasis) and metastatic colony formation (experimental metastasis) are performed as two distinct experiments. For primary tumor invasion, a bolus of cells is injected directly into the CAM stroma where it forms a large primary tumor similar to a mouse subcutaneous tumor. Control and screen-identified mutant cells show a differential ability to invade at the invading front of these tumors (**Fig. 2**). For the analysis of metastatic colonies, cells are injected IV and allowed to extravasate (as we demonstrated in Cell Rep. 2014 8:1558-70), similar to a tail vein injection of cancer cells in the mouse.

In the mouse model we only performed spontaneous metastasis experiments to assess the effect of the screen-identified genes on all stages of metastasis. Primary tumor growth was comparable between control and screen-identified mutant cells (**revised Supplementary Figs. 6 and 7**). We originally presented lung metastasis data using whole mount fluorescence imaging of the lungs and a sensitive alu-PCR method to detect overall metastatic burden. Indeed, inhibition of the screen-identified targets significantly decreased the dissemination of cancer cells from the primary tumor to the lungs. In the revised manuscript, we have performed additional experiments to characterize the lung metastases. We include representative images of mouse lung metastatic lesions (**new Supplementary Fig. 6c**).

6. “Why were only some genes evaluated in some human cancers? Are the ones shown in Figure 4 the only ones evaluated, or were the others examined but are not differentially expressed in primary and metastatic sites? Are the genes evaluated differentially expressed in primary and metastases in the avian or mouse models? Also, are any of the samples paired primary/metastasis?”

Author response: Good questions. We focused our analysis on the top screen-identified hits that showed significant difference in compactness index (CI) and also were novel genes represented by single shRNAs within the clones. For the data in the **revised Supplementary Figure 8**, we utilized the OncoPrint tool to query datasets that contained expression data from both primary and metastatic tumors (samples are matched). Only the datasets that show significant difference ($p < 0.05$) in expression of these hits in particular human cancers (primary tumor vs metastatic lesion) are shown. We plan to examine the expression of these genes between primary and metastatic sites in chicken and mouse however these experiments have not yet been performed.

7. *“There is no information on the human tissue sections in the methods. How many were analyzed? Staining procedures? Antibody validation? etc. Also, increased expression at the invasive edge is not apparent from the images, and quantification would strengthen this argument.”*

Author response: In the original version of manuscript we used only images acquired from Protein Atlas (5-10 patients per target). To address this concern, as well as a similar comment from reviewer #1, we have performed immunohistochemistry for Kif3b and SRPK1 on a prostate cancer progression cohort tissue microarray (TMA) from the University of Calgary (98 patients). Antibodies were validated by a board-certified Pathologist (new co-author Tarek Bismar) and the IHC protocol is now incorporated into the revised manuscript. As requested, we quantified the Kif3b and SRPK1 staining intensity in the invasive zones and it is included in the new version of the manuscript (**new Fig. 6**).

8. *“In the methods it states that at least 20 cells were tracked for each control and knockdown clone, but how many clones were evaluated for each? How were the 20 cells for each selected? There needs to be more description of how the velocity and displacement (i.e. productivity of cell migration) were calculated. Why are velocities reported as $\mu\text{m}^2/\text{sec}$? Why is the μm unit squared? Mean squared displacement would also be a valuable measurement to compare movement rates. Was displacement calculated over the entire track?”*

Author response: Cells were selected randomly and tracked using tools in the Improvisation Volocity tracking module. Definitions of track velocity and displacement rates as in Volocity software manual are included in the revised manuscript. Displacement was calculated over the entire track. We thank the reviewer for pointing out the error in **Figure 2** label and the current, corrected label is $10^{-2}\mu\text{m}/\text{sec}$.

9. *“Productivity of cell migration” is used throughout the text, but is never well defined, except in the figure legends. A more descriptive term or clearer definition is needed in the text.*

Author response: To address this, we've included a clearer definition in the text on page 7, line 13. A description of productivity/displacement rate measurement as it is used in the Volocity software is now added to “Material and Methods” section.

10. *“In a few instances the authors claim that their screening approach demonstrates a powerful tool to identify novel therapeutic targets. This is somewhat overstated, since there is no validation of whether the genes identified are actually good therapeutic targets.”*

Author response: To address this concern, we have changed the wording to “potential” therapeutic targets throughout the text. In our ongoing studies, we have performed experiments demonstrating that therapeutic inhibition of several of the screen-identified targets does indeed block metastasis, but since these will take some time to repeat/validate, we have decided to leave those experiments out of the current submission.

11. *“A description of panel f is missing from the figure legend for supplementary figure 3.”*

Author response: We added the missing description.

12. There is a typo in the label in figure 2a, bottom panel.

Author response: We corrected the typo.

In conclusion, we have produced a considerable amount of new data during the revision process, which has addressed all of concerns raised by both reviewers. These results have significantly improved our manuscript and we thank the reviewers for their constructive critiques of the manuscript. We hope that our revised manuscript is now suitable for publication in Nature Communications and look forward to hearing from you soon.

Reviewers' comments:

Reviewer #1 (Remarks to the Author):

The manuscript describes a high-throughput screening platform for quantitative identification of novel genes involved in cellular motility. In the revised manuscript, the authors have demonstrated the similarity in extracellular matrix between a chicken CAM and mouse lung tissue. This reviewer commends the authors for establishing a robust in vivo platform to replace in vitro Boyden chamber-based screens. Overall, the manuscript would benefit and could be further advanced by considering the following editorial comments.

1. Tumor cells, transfected with shRNA targeting Kif3b, show delayed tumor growth when compared to the scrambled control (supplementary figure 7a). It is recommended to include the data for both shRNA 1 and 2 in figures 3 and 4. Additionally, the authors should correct the legend of figure 4 to include 4e and 4f.
2. In figure 6, scale bars are missing for the IHC images.
3. In supplementary figures 4 and 5, the authors include data for shRNA 2, while describing shRNA 1 in the figure legends. This reviewer would strongly advise the authors to include the data for both shRNA 1 and 2 in all the figures to maintain consistency throughout the manuscript.
4. In supplementary figure 6, the authors need to replace the images for 6g and 6h.

Reviewer #2 (Remarks to the Author):

The authors have done a thorough job in addressing my previous concerns. They have included additional data that is convincing and addresses previous concerns. There are several improvements to the Materials and Methods section, with additional details now included. I am satisfied with the quality and extent of the revisions, and have no further concerns. This work represents an important advancement with an in vivo whole genome screen for motility and metastasis, and characterization of a top target from the screen (Kif3b), with validation in human cancer samples.

Response to Reviewer #1

1. *Tumor cells, transfected with shRNA targeting Kif3b, show delayed tumor growth when compared to the scrambled control (supplementary figure 7a). It is recommended to include the data for both shRNA 1 and 2 in figures 3 and 4. Additionally, the authors should correct the legend of figure 4 to include 4e and 4f.*

Author response: We have performed additional experiments and included data for both Kif3b shRNAs 1 and 2 in **revised Figures 3 and 4** and **newly added Supplementary Figure 4**. We have corrected the Figure 4 legend and manuscript text to better discriminate between shRNAs discovered in the process of screening (sh1) and independent shRNAs used for validation (sh2).

2. *In figure 6, scale bars are missing for the IHC images.*

Author response: Scale bars have been added to Figure 6.

3. *In supplementary figures 4 and 5, the authors include data for shRNA 2, while describing shRNA 1 in the figure legends. This reviewer would strongly advise the authors to include the data for both shRNA 1 and 2 in all the figures to maintain consistency throughout the manuscript.*

Author response: We have modified the figure legends to reflect the correct data. Because shRNA2 was used for validation of all of the screen hits in multiple cell lines, and because we observed consistent phenotypes using shRNA 1 and 2 in all of our in vitro and in vivo studies, we have opted to leave the

data in Supplementary Figures 4 and 5. We think that the new intravital imaging data provided in **revised Figures 3 and 4 and in Supplementary Figure 4** is sufficient to validate the role of Kif3b in invasion and metastasis.

4. In supplementary figure 6, the authors need to replace the images for 6g and 6h.

Author response: Improved images are provided in the revised Supplementary Figure 7.

I thank you for your time and hope you will now consider this manuscript suitable for publication.

REVIEWERS' COMMENTS:

Reviewer #1 (Remarks to the Author):

This reviewer would like to congratulate the authors for solidly establishing this robust quantitative in vivo screening platform.